# Molecular Docking of Lactoferrin with Apoptosis-Related Proteins Insights into Its Anticancer Mechanism

**DOI:** 10.3390/ijms26052023

**Published:** 2025-02-26

**Authors:** Lidia Esmeralda Angel-Lerma, Javier Carrillo-Campos, Luis Ignacio Siañez-Estrada, Tania Samanta Siqueiros-Cendón, Dyada Blanca León-Flores, Edward Alexander Espinoza-Sánchez, Sigifredo Arévalo-Gallegos, Blanca Flor Iglesias-Figueroa, Quintín Rascón-Cruz

**Affiliations:** 1Facultad de Ciencias Químicas, Universidad Autónoma de Chihuahua, Campus II Circuito Universitario s/n, Chihuahua 31125, Mexico; esme.angel.lerma@gmail.com (L.E.A.-L.); lsianez@uach.mx (L.I.S.-E.); tsiqueiros@uach.mx (T.S.S.-C.); p375694@uach.mx (D.B.L.-F.); eaespinoza@uach.mx (E.A.E.-S.); sareval@uach.mx (S.A.-G.); bfigueroa@uach.mx (B.F.I.-F.); 2Facultad de Zootecnia y Ecología, Universidad Autónoma de Chihuahua, Periférico Francisco R. Almada km 1, Chihuahua 31453, Mexico; jccarrillo@uach.mx

**Keywords:** Lactoferrin, apoptosis, molecular docking, XIAP, Caspase-3, cancer research

## Abstract

Human Lactoferrin (hLf), a multifunctional glycoprotein, has been analyzed through molecular docking to evaluate its role in apoptosis regulation and its potential as an anticancer agent. The docking results highlight XIAP (X-linked Inhibitor of Apoptosis Protein) and Caspase-3 as the most reliable targets, where hLf disrupts XIAP’s inhibition of Caspase-3 and Caspase-9, potentially restoring apoptotic signaling; hLf also stabilizes Caspase-3, enhancing its activation in intrinsic and extrinsic pathways. Weaker interactions were observed with Fas, Bcl-2, and Akt. hLf’s role in Fas signaling is likely due to expression upregulation rather than direct binding. In contrast, its binding to Bcl-2 may disrupt anti-apoptotic function, and its interaction with Akt suggests interference with pro-survival signaling. These findings suggest that hLf may promote apoptosis by enhancing caspase activation and modulating key apoptotic regulators, supporting its potential use in cancer treatment. However, further experimental validation is needed to confirm these interactions and their therapeutic implications.

## 1. Introduction

### 1.1. Cancer Overview

Cancer encompasses a diverse group of diseases originating in nearly any organ or tissue, characterized by uncontrolled cell growth, local tissue invasion, and metastasis, the leading cause of cancer-related deaths [1,2]. It is the second leading cause of mortality worldwide, with lung, breast, colorectal, prostate, and stomach cancer being the most prevalent [3]. In 2022, approximately 20 million new cases were diagnosed, with projections indicating a rise to 35 million by 2050 due to population growth [4].

Conventional cancer treatments, including chemotherapy and radiotherapy, face limitations such as non-specificity, toxicity, and the emergence of resistance, leading to increased recurrence rates and treatment inefficacy. These challenges underscore the need for targeted, less toxic therapeutic strategies [5,6,7].

### 1.2. Apoptosis and Cancer

Apoptosis, or programmed cell death, plays a crucial role in eliminating defective cells and maintaining cellular homeostasis. Dysregulation of apoptosis is a hallmark of cancer, often involving the overexpression of anti-apoptotic proteins (e.g., Bcl-2, Akt) and suppression of pro-apoptotic factors (e.g., Bax, Bim) [4]. Apoptosis occurs through two pathways: the intrinsic (mitochondrial) and extrinsic (death receptor-mediated) pathways, both culminating in caspase activation and cell death [8,9,10,11,12].

The intrinsic pathway responds to internal stimuli like DNA damage and oxidative stress, regulated by Bcl-2 family proteins. It leads to mitochondrial outer membrane permeabilization (MOMP), cytochrome c release, apoptosome formation, and Caspase-9 activation, ultimately triggering executioner caspases [13,14,15,16,17,18,19,20]. The extrinsic pathway is activated by death receptors such as Fas (*CD95*) and TRAIL-R1/R2, leading to Caspase-8 activation, which either directly activates executioner caspases or links to the intrinsic pathway via Bid cleavage [8,15,21]. Figure 1 provides an overview of these apoptotic pathways.

### 1.3. Evasion of Apoptosis in Cancer

Cancer cells employ multiple mechanisms to evade apoptosis, including *TP53* inactivation, upregulation of anti-apoptotic proteins (Bcl-2, Bcl-xL, Akt), and suppression of pro-apoptotic factors (Bax, Bim) [22]. The PI3K/AKT signaling pathway, frequently dysregulated in cancer, promotes survival and apoptosis resistance [22]. Additionally, tumors can downregulate death receptor expression, disrupting extrinsic apoptosis signaling [7,23,24].

This study focuses on eight key apoptotic proteins, Akt (*AKT1*), Bcl-2, p53, Fas (receptor), Caspase-3, Caspase-8, Caspase-9, and XIAP (X-linked Inhibitor of Apoptosis Protein), due to their critical roles in apoptosis regulation and cancer progression. Table 1 summarizes the central role of selective proteins in the apoptotic process and their implications for cancer.

### 1.4. Lactoferrin and Its Role in Cancer

Lactoferrin (Lf) is a multifunctional iron-binding glycoprotein present across mammalian species, with human and bovine Lf sharing 69% sequence identity [13,57,58]. It plays a role in iron homeostasis, antimicrobial defense, immune modulation, and cancer suppression [35,59,60]. Lf modulates apoptotic pathways by enhancing MOMP and interacting with p53 to regulate pro-apoptotic gene expression and DNA damage responses [13,58,61,62,63]. Additionally, Lf exhibits selective cytotoxicity against cancer cells and enhances chemotherapy efficacy [18,63,64,65].

This study investigates human Lactoferrin’s (hLf) role in apoptosis through its molecular interactions with Akt, Bcl-2, p53, Fas, Caspase-3, Caspase-8, Caspase-9, and XIAP. Structural prediction and molecular docking analyses were conducted to propose a potential mechanism of hLf’s pro-apoptotic activity.

## 2. Results

### 2.1. AlphaFold-Predicted Models of Human Lactoferrin, Caspase-3, Caspase-9, Caspase-8, Akt, Fas, Bcl-2, p53 and XIAP Proteins

The AlphaFold 3 server [66] was used to generate predicted structures and interactions between human Lactoferrin (hLf) and several proteins involved in apoptosis, including the Fas (receptor), p53, Caspase-9, Caspase-8, Caspase-3, Bcl-2, Akt (*AKT1*), and XIAP. The quality of these models was assessed using the Template Modeling (pTM) score for structural confidence, the interface pTM (ipTM) score for interaction prediction confidence (Figure 2), and the predicted aligned error (PAE) for positional confidence between chains. Lower PAE values indicate higher confidence. The fraction of disordered residues in the predicted complex was also evaluated to infer structural stability.

The hLf predicted structures showed high pTM scores (0.82–0.88), indicating strong confidence in its predicted structure. However, the confidence in its interactions with other proteins varied.

HLf-Caspase-9 and hLf-Caspase-8: These complexes showed the highest ipTM scores (0.34 and 0.25, respectively) with low PAE values (18.56–13.07 Å and 14.64–14.57 Å, respectively). These results indicate moderate confidence in the predicted interactions and relative protein positions. Additionally, good overall pTM scores (0.62 and 0.58, respectively) and high individual pTM scores for both proteins suggest these interactions may be relatively stable.

HLf-Caspase-3: Despite high pTM scores for hLf and Caspase-3 individually, the complex had a low ipTM score (0.14) and high PAE values (25.51 Å and 25.24 Å). This suggests low confidence in direct interaction. Notably, Caspase-3’s self-interaction showed a high ipTM score (0.75), indicating a strong dimerization tendency. The low fraction of disordered residues (0.05) in the predicted complex further highlights uncertainty regarding direct hLf-Caspase-3 interaction.

HLf-p53: Like hLf-Caspase-3, this complex showed a low ipTM score (0.17) and high PAE values (24.89 Å and 22.64 Å). Although the overall pTM score was moderate (0.6) and the individual pTM scores for Lf and p53 were high, these findings indicate uncertainty in the interaction between these proteins.

HLf-Bcl-2, hLf-Fas, and hLf-Akt: These complexes had low ipTM scores (0.11, 0.13, and 0.19, respectively) and high PAE values (approximately 29 Å for both), suggesting low confidence in their predicted interactions. Interestingly, Bcl-2 and Akt displayed relatively high ipTM scores for their self-interactions (0.61 and 0.67, respectively), suggesting a potential preference for homodimerization or interaction with other partners. The Fas receptor showed a low ipTM score (0.37), which may reflect structural uncertainty within the Fas model.

HLf-XIAP complex: The resulting model suggests that hLf and XIAP interact with a moderate level of confidence (ipTM score of 0.58) in the predicted interface. The overall model quality is reasonable, with an overall pTM score of 0.66 and no steric clashes detected. While most of the individual chains in the complex show high confidence (pLDDT > 90), some regions, particularly at the interface and in loop regions, have lower confidence scores (70 < pLDDT < 90).

All interaction analyses were performed using PDBsum*1* [67]. Key interaction residues were identified, and their contributions to hydrogen bonding, salt bridge formation, and non-bonded contacts were analyzed.

#### 2.1.1. HLf-Akt

The interaction analysis revealed eight hydrogen bonds, four salt bridges, and fifteen non-bonded contacts (Figure 3).

On hLf, the most frequently interacting residues included ARG31 (five interactions), ARG29 (four interactions), ARG25 (three interactions), ARG28 (three interactions), and LYS286 (two interactions). These residues exhibited versatility in forming hydrogen bonds, salt bridges, and non-bonded contacts, ensuring the stability of the hLf-Akt interface. Hydrogen bonds, crucial for specificity and stability, involve the residues ARG31, ARG29, GLN14, GLN22, ARG25, and ARG28, with ARG31 forming three such interactions. While salt bridges were less frequent compared to hydrogen bonds and non-bonded contacts, they played a critical role in electrostatic interactions, contributing to the stability of the predicted complexes. These interactions help stabilize the interface by providing long-range electrostatic attraction, which is particularly relevant for highly charged regions, with ARG29, ARG25, and LYS286 contributing prominently. Non-bonded contacts, the most abundant interaction type, ensured proximity and overall structural cohesion of the hLF-Akt interface. Initially, more than 15 non-bonded contacts were identified. After averaging interactions involving the same residue pairs, this was refined to eleven averages and four unique contacts; ARG28 and ARG31 were predominant residues.

On Akt, the most interacting residues included ARG243 (four interactions), GLU278 (three interactions), and GLN352 (three interactions). Hydrogen bonds were formed by SER240, GLU278, TYR340 (2twohydrogen bonds), LEU347, ARG346, GLU440, and GLU441, reinforcing the stability of the complex. Salt bridges were mediated by GLU278 and GLU314, which provided essential electrostatic stabilization. Non-bonded contacts were the most frequent interaction type, with ARG243, GLU278, and GLN352 being notable contributors. The interaction analysis revealed eight hydrogen bonds, four salt bridges, and fifteen non-bonded contacts (Figure 3).

#### 2.1.2. HLf-Bcl-2

The molecular docking analysis between hLf and Bcl-2 revealed three hydrogen bonds, five salt bridges, and seven non-bonded contacts (Figure 4).

On hLf, the most frequently interacting residues included LYS74 (three interactions), ALA71 (two interactions), PRO72 (two interactions), ARG333 (two interactions), and GLU353 (two interactions). Hydrogen bonds were rare but critical for ensuring specificity with LYS74 (two bonds) and GLU353. GLU52, LYS74, ARG333, ARG345, and GLU353 mediated the salt bridges, which enhanced electrostatic stabilization. Non-bonded contacts were the most abundant interaction type, playing a key role in maintaining the overall structural stability of the complex (six average and one unique non-bonded contact). The residues ALA71 and PRO72 were notably involved in this interaction.

On Bcl-2, the most interacting residues are ARG106 (two interactions), SER116 (two interactions), GLU160 (two interactions), ARG164 (two interactions), and GLU165 (two interactions). These residues contributed significantly to the hLf-Bcl-2 interaction through their involvement in hydrogen bonds, salt bridges, and non-bonded contacts. Hydrogen bonds involved the residues ARG106, SER116, and GLU160 stabilizing the interface, while salt bridges, crucial for maintaining electrostatic complementarity, were mediated by GLU29, ARG106, HIS120, GLU160, and GLU165. Non-bonded contacts played a pivotal role, with residues such as ARG164 forming close associations with hLf residue. The diverse interactions and residues underscore the essential role of hLf residues in forming stable and specific interactions with Bcl-2.

#### 2.1.3. HLf-Caspase-3

The molecular docking analysis between hLf and Caspase-3 revealed 12 hydrogen bonds, 3 salt bridges, and 31 non-bonded contacts (Figure 5). On hLf, the most frequently interacting residues included GLN14 (seven interactions), LYS39 (five interactions), GLN187 (five interactions), THR11 (three interactions), THR18 (three interactions), PHE21 (three interactions), ARG40 (three interactions), SER186 (three interactions), and GLN296 (three interactions). These residues played a key role in stabilizing the hLf-Caspase-3 complex. Hydrogen bonds were primarily mediated by THR11, GLN14, THR18, and GLN187, each forming two such interactions. Salt bridges involved ARG40, ASP41, and ARG54. Non-bonded contacts, the most abundant interaction type (29 average and 2 unique non-bonded contacts), helped maintain proximity and structural cohesion. Notably, GLN14 and LYS39 each formed five non-bonded contacts, while GLN187 formed three, emphasizing their pivotal roles in ensuring interface stability.

On Caspase-3, the principal interacting residues included THR166 (six interactions), GLU167 (six interactions), LEU168 (four interactions), and THR255 and GLU123 (four interactions). These residues contributed substantially to the hLf-Caspase-3 interaction through a combination of interactions. Hydrogen bonds were predominantly mediated by THR166, GLU167, LEU168, and SER251, increasing the complex’s stability through specific interactions. Salt bridges, crucial for electrostatic complementarity, were formed by HIS121, GLU123, and ASP135. Non-bonded contacts were most frequently observed for THR166, GLU167, THR255, and GLU123, highlighting their role in maintaining the integrity of the protein-protein interface.

#### 2.1.4. HLf-Caspase-8

The molecular docking analysis between hLf and Caspase-8 revealed three hydrogen bonds, seven salt bridges, and fourteen non-bonded contacts (Figure 6).

On hLf, the most frequently interacting residues included ARG4 (five interactions), ARG28 (four interactions), ARG5 (three interactions), ARG2 (two interactions), ARG25 (two interactions), and ARG29 (two interactions). These residues exhibited adaptability in forming various interaction types, supporting the hLf-Caspase-8 interface. Hydrogen bonds, though few, were primarily mediated by ARG4, ARG5, and ARG28. Salt bridges, crucial for electrostatic stabilization, involved ARG4, ARG25, ARG28, ARG29, and ARG31. Non-bonded contacts, the most frequent interaction type (nine averaged and five unique non-bonded contacts), ensured proximity and structural cohesion; the most repeatedly interacting residues were ARG2, ARG4, ARG5, and ARG28.

On Caspase-8, hydrogen bonds were observed with ASP18, TYR78, and TYR79, while salt bridges were mediated by ASP9, GLU12, ASP15, GLU17, ASP18, GLU87, and GLU110. Non-bonded contacts played a pivotal role in maintaining the structural integrity of the interface, involving 14 different residues of Caspase 8.

#### 2.1.5. HLf-Caspase-9

The molecular docking analysis between hLf and Caspase-9 revealed 15 hydrogen bonds, 6 salt bridges, and 15 non-bonded contacts (Figure 7). On hLf, the most frequently interacting residues included ARG3 (four interactions), LYS39 (four interactions), ARG40 (three interactions), GLN24 (two interactions), GLU67 (two interactions), ARG122 (two interactions), ASN127 (two interactions), and LYS302 (two interactions). These residues are important in supporting the stability of the hLf-Caspase-9 complex. Hydrogen bonds were predominantly mediated by ARG3, LYS39, and ARG40, each contributing two bonds. Salt bridges were formed by ASP23, ASP27, GLU41, ASP79, ASP83, and ASP327. Non-bonded contacts, involving residues such as ARG3 and LYS39, were the most abundant interaction type (seven averaged and nine unique non-bonded contacts), ensuring proximity and structural integrity. On Caspase-9, key residues included ASP79 (five interactions), ASP330 (four interactions), ASP27 (three interactions), ASP327 (three interactions), and GLN328 (three interactions). These residues exhibited the most interactions and stabilized the LF-Caspase-9 interface through a combination of interactions. Hydrogen bonds were formed with ASP27, PRO37, ASP79 SER242, ASP330, and GLN328, while non-bonded contacts involved residues such as ASP79 and ASP327, emphasizing their role in maintaining the structural cohesion of the complex.

#### 2.1.6. HLf-p53

The molecular docking analysis between hLf and p53 revealed 10 hydrogen bonds, 3 salt bridges, and 18 non-bonded contacts (Figure 8).

On hLf, the most frequently interacting residues included ARG3 (three interactions), ARG5 (three interactions), GLU52 (three interactions), and LYS74 (three interactions). These residues contribute to the stability of the hLf-p53 interface. Hydrogen bonds were mediated by ARG121, ARG3, ARG5, SER6, GLN8, GLU53, ASN53, LYS74, and GLN330. Salt bridges involving ARG5, GLU52, and LYS74 were less frequent but provided critical electrostatic stabilization. Non-bonded contacts, the most abundant interaction type (5 averaged and 13 unique non-bonded contacts), ensured structural proximity and reinforced the complex’s stability; the predominant residues were ARG3, ARG5, and ASN53. On p53, the primary interacting residues included SER95 (three interactions), TYR103 (three interactions), and GLU336 (three interactions). These residues contribute significantly to the stability of hLf-p53 interaction through the different interaction types. Hydrogen bonds were formed with SER95, TYR103, ASN210, ARG267, GLU336, GLU339, and GLU346, while salt bridges involved the residues GLU258, ARG267, and GLU339. Non-bonded contacts ensured the structural integrity of the interface; the prominent residues are GLU336, LEU350, TYR103, and ASN263.

#### 2.1.7. HLf-Fas

The molecular docking analysis between hLf and Fas revealed three hydrogen bonds, three salt bridges, and eleven non-bonded contacts (Figure 9).

On hLf, the most frequently interacting residues included ARG25 (three interactions), PHE21 (two interactions), GLN45 (two interactions), and GLU179 (two interactions). GLN24, GLN45, and GLU179 predominantly mediated hydrogen bonds, while salt bridges were primarily formed by ARG25, ARG121, and LYS297, contributing to electrostatic stabilization. In non-bonded contacts, the most common interaction type (four averaged and seven unique non-bonded contacts) ensured structural cohesion and proximity; PHE21 and ARG25 were the noticeable residues.

On Fas, the key interacting residues included ASP55 (four interactions), GLN283 (three interactions), and HIS54 (two interactions). HIS53, THR147, and GLN283 formed hydrogen bonds, while salt bridges involving ASP55, GLU256, and ASP265 added an additional layer of stabilization through electrostatic interaction. In non-bonded contacts, ASP55 and GLN283 contributed the most to stabilizing the interface.

#### 2.1.8. HLf-XIAP

The PDBsum*1* analysis reveals that hLf and XIAP interact through a diverse combination of non-bonded contacts, hydrogen bonds, and a salt bridge (Figure 10). The interaction involves a range of atoms, with distances between interacting atoms spanning from 2.68 to 3.9 Å, averaging at 3.53 Å. Notably, THR274, PHE270, ILE276, GLY273, and THR271 from hLf and ASP175, HIS178, GLY293, and PRO212 from XIAP participate in multiple non-bonded contacts, underscoring their potential role in stabilizing the interaction.

Particularly, ASP163 from hLf stands out by engaging in all three interaction types, highlighting its multifaceted contribution to the complex formation. The combination of interaction types and the proximity of the involved atoms strongly suggest a stable complex between hLf and XIAP.

### 2.2. Interaction Profile of hLf with Apoptotic Proteins

The docking analysis revealed a distinct pattern of interactions between hLf and apoptotic proteins, with significant clustering within specific structural regions of the protein. As shown in Figure 11, the highest density of interactions was observed within the Lactoferricin (Lfcin) region (residues 1–47), a key segment of the N-lobe (residues 1–333). This region exhibited the most substantial interaction frequency with apoptotic proteins, suggesting its primary role in mediating protein-protein interactions.

Beyond Lfcin, additional interactions were distributed across the N-lobe, albeit with lower frequency, while interactions in the Connecting Helix (334–344) and C-lobe (345–691) were minimal. These findings suggest that the N-terminal domain, particularly the Lfcin segment, plays a dominant role in facilitating apoptotic protein binding.

This interaction pattern aligns with previous reports indicating that the Lfcin region harbors key functional residues involved in antimicrobial and immunomodulatory activities. Its high binding propensity with apoptotic proteins further supports its potential pro-apoptotic function. The Connecting Helix and C-lobe regions exhibited significantly fewer interactions, suggesting a lesser contribution to apoptotic protein binding.

Overall, these results highlight the importance of Lfcin in mediating interactions with apoptotic regulators, positioning it as a key functional domain for Lactoferrin’s apoptotic activity. Further experimental validation is needed to elucidate these interactions’ structural basis and biological relevance.

#### Negative Control Docking Outcomes

The docking metrics of the negative control between Caspase-8 and Ovalbumin showed an iPTM score of 0.12 (Appendix A), with PAE values ranging from 27.57 Å to 27.90 Å (Appendix A). The PAE Min and PAE Max values were the highest among the tested complexes (Appendix A), and the pTM score indicated low structural stability for this complex (Appendix A).

## 3. Discussion

### 3.1. HLf and Fas

Bovine Lactoferrin (bLf) has been shown to activate apoptosis through the Fas pathway. Specifically, bLf has been shown to significantly increase Fas protein expression in the proximal colon mucosa of rats treated with azoxymethane, a potent carcinogen. This elevated Fas expression in the colon mucosa was accompanied by an increase in apoptotic cells and the activation of caspase-8 and caspase-3, which are key downstream effectors of the Fas pathway (extrinsic apoptotic pathway) [68,69]. While evidence supports Lf’s role in promoting apoptosis via Fas expression, we aimed to investigate whether hLf could directly interact as a ligand to activate Fas.

The docking analysis between hLf and Fas reveals that the interacting residues in Fas are located at positions 52–55, 57, 145, 147, 256, 283, 264, and 265, all of which are situated outside the CRD2 domain (positions 85–127). Only two residues are located within the CRD3 domain (positions 129–149) [70]. CRD2 and CRD3 are domains in the extracellular region of Fas, primarily involved in ligand binding, with CRD2 playing a key role in ligand binding and CRD3 contributing to ligand specificity [71,72,73]. Despite the interaction with two residues in the CRD3 domain, it has been demonstrated that residues ARG86 and ARG87, located within the CRD2 domain, are essential for ligand binding [71,74]. This suggests that hLf does not act as a ligand for Fas. These findings indicate that hLf’s pro-apoptotic role in enhancing Fas activity is mainly due to the increased expression of Fas, as previously reported. However, it is unlikely to function as a Fas ligand.

### 3.2. HLf and Caspases

Lactoferrin has emerged as an intriguing modulator of apoptosis. Central to this modulation is its interaction with caspases 3, 8, and 9, key executioners of apoptosis [18].

The effect of Lactoferrin on caspases 3, 8, and 9 involves a complex interplay influenced by various factors, including the specific type of Lactoferrin (bovine Lactoferrin or bovine Lactoferricin), the cell type involved, and the cellular environment. All studies have been conducted using bovine Lactoferrin (bLf) or its derivative bovine Lactoferricin (Lfcin-B).

For instance, in Jurkat leukemia T cells, treatment with bLf leads to elevated levels of active caspases 3 and 9 [9,75]. Another study showed that after oral administration of bLf, there was a significant increase in the active forms of Caspase-8 and Caspase-3 [68]. In breast cancer cells, it has been shown that cleavage of Caspases 8 and 9 was increased when treated with bLf [10]. Similarly to SAS oral cancer cells, bLf also induces Caspase-3 activation [76]. In lung cancer H460 cells, the peptide Lactoferricin-B (Lfcin-B) stimulated Caspases 3 and 9 [77]. Lfcin-B also promotes cleavage of Caspases 3, 8, and 9 when treated in the human gastric cell line AGS [78].

The evidence that Lf generally promotes apoptosis through caspase activation indicates that Lf is potentially interacting with these proteases.

The protein-protein docking results of hLf with Caspase-3 evidence the greatest number of interactions compared with the other two caspases. The catalytic domain of Caspase-3, particularly the two catalytic residues Cys163 and His121 [79], interacts with hLf (Figure 5), implying that direct interaction with hLf could potentially inhibit the activity of this enzyme. This was not the case for Caspase 8 and 9.

Caspase 9 has some important residues for its function, a catalytic Cys287 [80], Phe404 that plays a crucial role in activation [81], Arg180 and Arg191 that, when mutated, impairs the ability of Caspase-9 to interact with the protein Apaf-1, which is necessary for its activation [82,83], Tyr153 that is phosphorylated by c-Abl tyrosine kinase in response to DNA damage, which enhances Caspase 9 auto processing [84], Thr125, a phosphorylation site targeted by multiple kinases like ERK2, DYRK1A, CDK1/cyclin B1, and P38α MAPK, inhibiting Caspase 9 processing, Ser196, which is a phosphorylation site targeted by the protein kinase Akt inhibiting Caspase 9 processing in vitro [85], and Ser144 that is phosphorylated by PKCξ, inhibiting Caspase-9 processing [86]. All these residues were not interacting with hLf (Figure 7), and the regions where these residues are allocated are not affected by the interaction with hLf, indicating that all the different protein-protein interactions necessary for Caspase-9 physiological role could, in principle, be accomplished even with the presence of hLf.

Two potentially important residues of Caspase-9 interact with hLf: Asp315 and Asp330; these are cleavage sites that generate neoepitopes. These residues are targeted by different proteases, resulting in distinct forms of cleaved Caspase-9 with varying susceptibility to regulation. Asp315 is the primary autocatalytic cleavage site for Caspase-9. When Caspase-9 dimerizes within the apoptosome, it gains the ability to cleave itself at Asp315. This auto-processing generates a cleaved form of Caspase-9 (cl-caspase-9) that exposes the Asp315 neoepitope. Asp330, on the other hand, is targeted by Caspase-3. Caspase-3, an executioner caspase-activated downstream of Caspase-9, cleaves Caspase-9 at D330, producing a distinct cl-caspase-9 with the D330 neoepitope [87].

Asp315 cl-caspase-9 and Asp330 cl-caspase-9 are fully active proteases, meaning they can cleave their downstream targets and propagate the apoptotic cascade. However, they differ in their sensitivity to inhibition by XIAP [87]. Based on our docking results, it appears that the interaction between these two residues with hLf could potentially risk the activity of Caspase-9. While Caspase-9 processing was initially thought to be essential for its activation, more recent research indicates that it primarily regulates the duration of apoptosome activity rather than directly activating Caspase-9 [88]. Therefore, direct binding of hLf with Caspase-9 could result in allosteric interactions where the binding induces a conformational change that enhances protease activity or facilitates its activation by other factors.

Caspase-8 has important residues for its function: the catalytic Cys360 [89], Asp374, and Asp384. These aspartic acid residues are the cleavage sites between the p18 and p12 subunits of Caspase-8 cleavage at these sites and are crucial for the maturation and full activation of Caspase-8 [90]. These residues were unaffected by hLf interaction (Figure 6), indicating that Caspase-8 could well undergo activation. Interestingly, hLf interacts with residues from the N-terminal region of Caspase-8 (1–110), which in this region is located the DED1 domain (1–80) and the DED2 domain (100–177) [90]; both domains are important for stability of the procaspase-8 before activation and for regulation of the Caspase-8 activity. For example, c-FLIPS binds to the DED filament, inhibiting caspase-8 activation and filament elongation [90]. The binding of hLf in this region could serve both as a stabilizer of procaspase-8 and facilitate activation, as well as being an inhibitor of the c-FLIPS protein binding, therefore promoting apoptosis.

### 3.3. HLf and p53

Lactoferrin (Lf) has demonstrated the ability to modulate p53 by upregulating *TP53* expression through NF-κB activation and promoting the transcription of p53-dependent genes [13,62]. It has also been shown to activate p53 via kinase-mediated phosphorylation, enhancing its role in cell cycle arrest and apoptosis [61,91]. While the data do not explicitly confirm that Lf directly binds to the p53 protein, they strongly suggest an interplay between Lf and p53.

The docking analysis revealed that hLf interacts with several residues of p53. Some of these residues are located within important functional domains of p53, including the transactivation domain (TAD), DNA-binding domain (DBD), and tetramerization domain (TD) [13,91]. However, the interactions do not strongly overlap with residues critical for p53’s primary activities. In the TAD, hLf binds to SER90 and SER95, which may stabilize the domain or enhance kinase interactions. However, it does not interact with key phosphorylation sites such as SER15 or SER20, limiting its potential influence on transcriptional activation [61].

In the DBD, the interaction with ARG267 could stabilize the DNA-binding region. However, the absence of interactions with essential DNA-contact residues like ARG248 and ARG273 reduces the likelihood of significant effects on p53’s DNA-binding capacity [13]. The most notable interactions are observed in the TD, where hLf binds residues such as GLU339 and LEU350, suggesting a potential role in stabilizing p53 tetramerization, which is essential for its functional assembly.

### 3.4. HLf and Akt

Our docking experiments between hLf and Akt have yielded interesting insights into the potential interaction between these two proteins. The results suggest that hLf binds to a specific region on Akt without interfering with several key kinase functions. Specifically, our docking model indicates that hLf does not obstruct the ATP-binding site or the PDK1-interacting fragment (PIF) pocket. This suggests that hLf binding may not directly inhibit the Akt activation and ATP binding process.

However, we observed that hLf binds to a region on Akt where three key glutamate residues, Glu234, Glu278, and Glu341, are located (Figure 3). As demonstrated in previous studies, these residues are crucial for substrate binding and specificity [92]. They interact with specific residues in the substrate, primarily arginine residues at the P(-3) and P(-5) positions, which are characteristic of many Akt substrates; for example, the Akt phosphorylation site on the pro-apoptotic protein BAD (Ser136) has arginine residues at the P(-3) and P(-5) positions and a threonine residue at the P(-2) position. Similarly, the Akt phosphorylation site on the Forehead transcription factor FKHR (Ser253) also has arginine residues at the P(-3) and P(-5) positions [92]. The interaction between these glutamate residues and the substrate arginine residues involves the formation of salt bridges, which are strong electrostatic interactions that contribute significantly to the binding energy. Glu236 interacts with the arginine at the P(-3) position, while Glu279 and Glu342 interact with the arginine at the P(-5) position.

The binding of hLf to this region may disrupt the formation of these salt bridges, potentially interfering with Akt’s ability to bind and phosphorylate its substrates. This interference could have downstream effects on Akt signaling pathways, as the phosphorylation of Akt substrates is essential for various cellular processes, including cell growth, survival, and metabolism [28].

Further investigation is needed to confirm the functional implications of hLf binding to this region. It would be interesting to explore whether hLf can modulate Akt signaling by altering substrate specificity or competing with other regulatory proteins interacting with this region.

### 3.5. HLf and Bcl-2

Due to its critical role in cancer progression, Bcl-2 is a well-established target for cancer therapy [32,34]. Venetoclax, a selective Bcl-2 inhibitor, has been approved by the FDA to treat certain cancers, including chronic lymphocytic and acute myeloid leukemia [93]. In addition to venetoclax, other inhibitors such as navitoclax, sonrotoclax, and APG-2575 are currently in clinical trials or under investigation. [94,95,96]. However, using these inhibitors faces challenges, including the development of resistance and adverse effects [95]. All these inhibitors share a common mechanism: binding to the BH3-binding groove of Bcl-2 [93]. This interaction displaces pro-apoptotic proteins, which subsequently initiate mitochondrial outer membrane permeabilization, leading to the release of cytochrome c and apoptosis [32].

The docking results of hLf and Bcl-2 indicate that hLf is unlikely to bind directly to the BH3 groove of Bcl-2. Key residues critical for the binding of inhibitors to the BH3 groove, such as MET115, VAL133, ASP103, and GLU152, were not involved in the interaction with hLf [95]. However, the proximity of residues interacting with hLf to these critical BH3 groove residues suggests potential indirect effects. For instance, Asp103 is near Arg106, and Met115 is close to Ser116, Ser117, and His120, raising the possibility that hLf binding could induce allosteric modulation or conformational changes in Bcl-2.

These interactions may alter the structural or functional dynamics of the BH3 groove, potentially impacting its ability to bind pro-apoptotic proteins. This hypothesis highlights the need for further experimental validation to explore these interactions’ biological significance and potential therapeutic implications.

### 3.6. HLf and XIAP

The docking results between hLf and XIAP highlight a significant interaction involving the BIR2 and BIR3 domains of XIAP. These domains play essential roles in apoptosis inhibition and cancer progression. The analysis provides valuable insights into the structural basis of this interaction and its potential implications for XIAP’s function and therapeutic targeting.

The surface representation of the docked complex reveals that hLf primarily interacts with the BIR2 and BIR3 domains of XIAP (Figure 10). The BIR2 domain is critical for inhibiting Caspase-3 and Caspase-7, key executioners of the apoptotic pathway [97,98]. The observed binding interface between hLf and the BIR2 domain suggests potential disruption of these interactions, which could alleviate XIAP’s inhibition of apoptosis. Specifically, the spatial orientation of hLf within the complex may interfere with the binding of caspases to the IAP-binding motifs (IBMs) of the BIR2 domain.

Similarly, the interaction with the BIR3 domain is particularly intriguing, given its role in inhibiting Caspase-9 activity in the mitochondrial pathway of apoptosis [99]. HLf’s binding could potentially compete with Caspase-9 or SMAC/DIABLO [99] to access the BIR3 domain, thereby promoting apoptosis. Disrupting the BIR3 domain’s interaction with Caspase-9 or its antagonists could further amplify apoptotic signaling in cancer cells.

The dual interaction of hLf with both the BIR2 and BIR3 domains highlights its potential as a modulator of XIAP’s anti-apoptotic activity [100]. Targeting these domains, hLf may enhance the apoptotic response, particularly in cancer cells where XIAP is often overexpressed [54,101,102,103]. Furthermore, the binding of hLf to the BIR3 domain may facilitate the release of SMAC/DIABLO or ARTS, endogenous antagonists of XIAP, further promoting apoptosis.

The observed interactions suggest that hLf could serve as a lead compound or a scaffold for developing novel therapeutic agents targeting XIAP. Given the critical role of XIAP in cancer cell survival and its association with chemoresistance, disrupting its interactions with caspases and other regulatory proteins presents a promising strategy for cancer therapy. The binding of hLf to the BIR2 and BIR3 domains could potentially restore apoptotic pathways, making it a candidate for combination therapies with other chemotherapeutic agents.

The molecular docking results summarized in Figure 12 highlight hLf’s role in apoptosis modulation through interactions with key regulators in both the extrinsic and intrinsic pathways, as well as crosstalk amplification mechanisms.

In the extrinsic pathway, hLf enhances Fas expression, leading to Caspase-8 and Caspase-3 activation, and stabilizes Caspase-8 while inhibiting c-FLIPS, reinforcing apoptotic signaling.

In the intrinsic pathway, hLf stabilizes p53, enhances its tumor suppressor function, disrupts Bcl-2, and interacts with Caspase-9 cleavage sites, promoting its activation.

Additionally, hLf amplifies apoptosis crosstalk by:Potentially disrupting Akt substrate binding, affecting survival signaling.Interacting with XIAP, inhibiting suppression of Caspase-3 and Caspase-9.Stabilizing Caspase-3, ensuring efficient execution of apoptosis.

These findings suggest hLf as a pro-apoptotic regulator with potential therapeutic applications in cancer and immune regulation.

### 3.7. Negative Control Docking

The negative control docking experiment using Caspase-8 and Ovalbumin confirmed the specificity of AlphaFold 3 predictions. Low ipTM scores and high PAE values indicated a low likelihood of non-specific interactions, reinforcing the robustness of the model.

To further validate these findings, biochemical binding assays or molecular dynamics simulations could provide deeper insights into binding stability and functional relevance. The results support the reliability of the docking predictions, ensuring that interactions observed are biologically meaningful rather than artifacts of the computational process.

## 4. Materials and Methods

### 4.1. Protein Sequence Retrieval

The amino acid sequences of human Lactoferrin (hLf) and apoptosis-related proteins (Akt, Bcl-2, p53, Fas, Caspase-3, Caspase-8, Caspase-9, and XIAP) were obtained from the UniProtKB database (https://www.uniprot.org/) to ensure high-quality, manually curated, and experimentally validated data. When available, structural data were retrieved from the Protein Data Bank (PDB) (https://www.rcsb.org/). The following UniProtKB entries were used: hLf (B7ZAL5), Akt (P31749), Caspase-3 (P42574), Bcl-2 (P10415), Caspase-8 (Q14790), p53 (P04637), Fas (P25445), Caspase-9 (P55211), and XIAP (P98170).

### 4.2. Structure Prediction and Molecular Docking

To predict protein structures and interactions, we used AlphaFold 3 [66], which generates de novo structural predictions using deep-learning-based methodologies. The interactions between hLf and apoptosis-related proteins were modeled using AlphaFold’s multimer module, which predicts complex structures based on co-evolutionary data and deep-learning algorithms rather than traditional template-based homology modeling.

For each protein-protein interaction, AlphaFold 3 generated five models, and the highest-ranked model based on the ipTM score was selected for further analysis.

Validation of Structural Predictions

The quality of the AlphaFold 3-predicted models was assessed using the following confidence metrics:Predicted Template Modeling (pTM) score: Measures the reliability of the overall structural prediction, where values close to 1 indicate high confidence.Interface pTM (ipTM) score: Evaluates the confidence in the predicted binding interface of the protein-protein complex.Predicted Aligned Error (PAE): Quantifies the relative positional confidence between residues across the protein complex. Lower PAE values indicate greater confidence in the predicted spatial relationships.

Unlike homology-modeled structures, which rely on experimental templates and require Ramachandran plot validation, AlphaFold 3 integrates steric, torsional, and conformational constraints directly into its predictions. Therefore, Ramachandran plots provide limited additional insight for AlphaFold 3-generated models. Instead, we relied on pLDDT, pTM, ipTM, and PAE metrics, which have been demonstrated to correlate well with experimentally validated structures.

### 4.3. Protein-Protein Interaction Analysis

The top-ranked docked structures were subjected to interaction analysis using PDBsum*1* [67], which provided a detailed breakdown of the interactions.

Hydrogen bonds: Donor-acceptor interactions with distances calculated.Salt bridges: Electrostatic interactions between charged residues.Non-bonded contacts: Van der Waals and hydrophobic interactions.

All protein visualization images were generated using ChimeraX [104]. The selection of the best docking model was based on the ipTM score, PAE values, and visual inspection of key interface residues.

### 4.4. Negative Control Docking Experiment

To assess the specificity of docking predictions, a negative control docking experiment was performed using Caspase-8 and Ovalbumin, a protein with no reported interactions with apoptotic proteins. This test was designed to determine whether AlphaFold 3 would predict non-specific interactions between two unrelated proteins. The docking results showed significantly lower ipTM scores and higher PAE values compared to biologically relevant interactions. Detailed results, including numerical values and interaction analysis, are provided in the Appendix A.

## 5. Conclusions

This study highlights the molecular interactions between human Lactoferrin and key apoptotic proteins, offering insights into its potential anticancer mechanisms. Among the interactions analyzed, hLf’s ability to target XIAP and Caspase-3 emerges as the most supported and significant.

HLf’s interaction with XIAP is critical for disrupting XIAP’s inhibitory effects on Caspase-3 and Caspase-9, restoring caspase activity and amplifying apoptotic signaling. Additionally, hLf stabilizes and promotes the activation of Caspase-3, a key executioner in the apoptotic cascade. While hLf indirectly enhances Fas-mediated apoptosis and interacts with Bcl-2, these interactions are less supported than XIAP and Caspase-3.

Although molecular docking provides valuable insights, experimental validation is essential to confirm these findings and establish human hLf’s apoptotic role. Key experimental approaches include protein interaction studies (e.g., co-immunoprecipitation) to confirm direct binding, caspase activity assays to validate functional effects, and cell-based apoptosis assays to evaluate its physiological relevance. Additionally, gene expression analysis and XIAP inhibition assays can further elucidate its impact on apoptotic pathways.

The negative control docking experiment, using Caspase-8 and Ovalbumin, confirmed the specificity of the docking predictions, supporting the robustness of the computational approach.

Combining computational predictions with experimental validation will provide a comprehensive understanding of hLf’s therapeutic potential in modulating apoptosis and overcoming cancer-related resistance mechanisms.

## Figures and Tables

**Figure 1 ijms-26-02023-f001:**
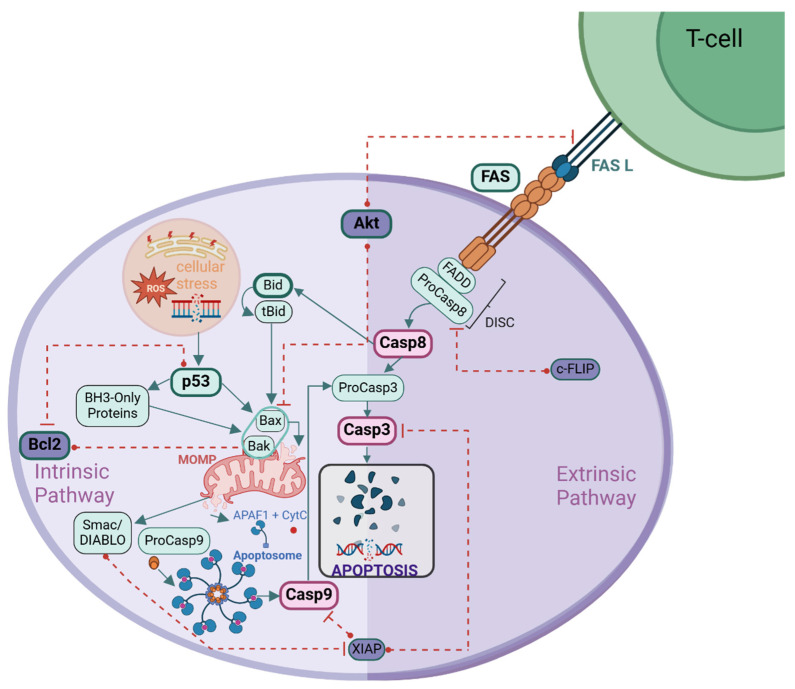
Intrinsic and Extrinsic Apoptotic Pathways. The intrinsic pathway is triggered by cellular stress, activating p53, which promotes Bax/Bak-mediated MOMP. This leads to cytochrome c (CytC) release, apoptosome formation, Caspase-9 activation, and subsequent Caspase-3-mediated apoptosis. Bcl-2 inhibits this process, while Smac/DIABLO counteracts XIAP to allow caspase activation. The extrinsic pathway is initiated by Fas Ligand (FASL) binding to Fas receptor (FAS), leading to Caspase-8 activation via the DISC (Death-Inducing Signaling Complex). Caspase-8 directly activates Caspase-3 or cleaves Bid into tBid, linking extrinsic and intrinsic pathways. Akt promotes survival by inhibiting apoptotic factors, while XIAP and c-FLIP suppress caspase activation, regulating apoptosis. Dashed red lines indicate inhibitory interactions, while solid arrows represent activation or progression in apoptotic signaling.

**Figure 2 ijms-26-02023-f002:**
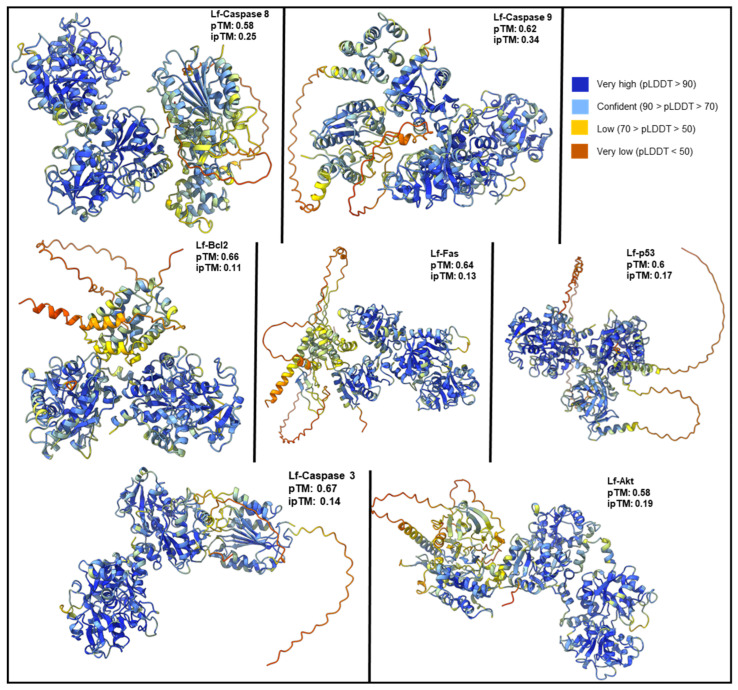
Predicted structures of hLf in complex with apoptotic proteins. AlphaFold3-generated models of hLf in complex with Caspase-8, Bcl-2, p53, AKT, Caspase-9, and Fas protein. The coloring of the protein structures corresponds to the predicted Local Distance Difference Test (pLDDT) score. The predicted Template Modeling (pTM) score, reflecting confidence in the overall structural prediction, and the interface pTM (ipTM) score, reflecting confidence in the interaction interface prediction, are indicated for each complex.

**Figure 3 ijms-26-02023-f003:**
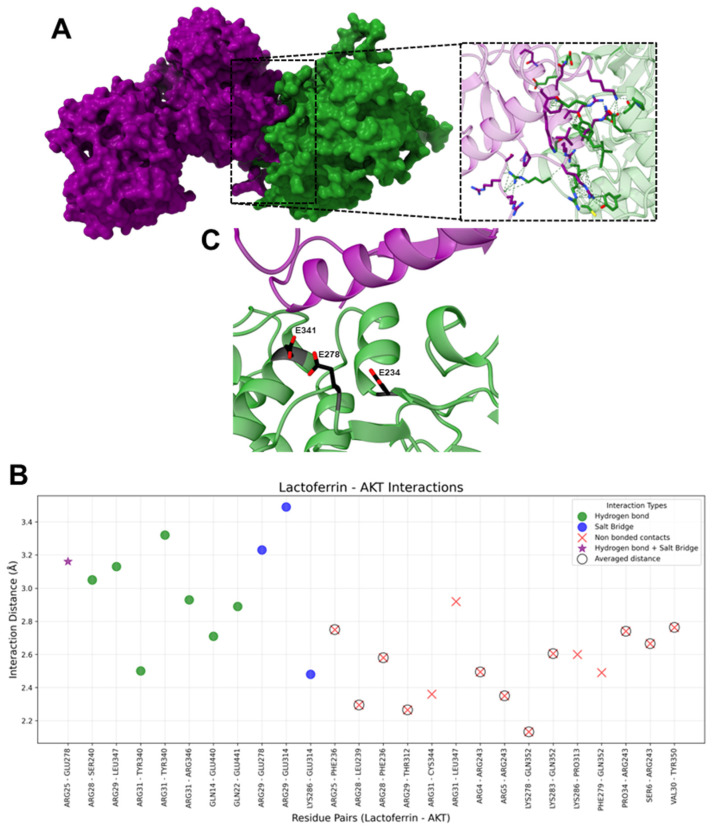
Interactions between hLf and AKT. (**A**) Molecular surface representation of the hLf-Akt complex. HLf is shown in purple and Akt is shown in green, highlighting the interface where interactions occur. The zoomed-in view displays the residues involved interacting in the complex, with green dashed lines representing contacts between residues. (**B**) The scatter plot illustrates interaction distances (Å) for residue pairs involved in Salt Bridges (blue), Hydrogen Bonds (green), Non-Bonded Contacts (red), and Hydrogen bond + Salt Bridge (purple star). Residue pairs with averaged distances for “Non-Bonded Contacts” (i.e., residue pairs that presented more than one “Non-Bonded Contact” interaction) are marked with hollow black circles around red “X” markers. The *x*-axis represents hLf and AKT residue pairs, and the *y*-axis represents the interaction distance in Å. (**C**) Ribbon representation of the hLf-Akt complex. HLf is shown in purple and Akt is shown in green. The three aspartic residues involved in substrate binding and recognition are shown as stick representations, with carbon atoms in black and oxygen atoms in red.

**Figure 4 ijms-26-02023-f004:**
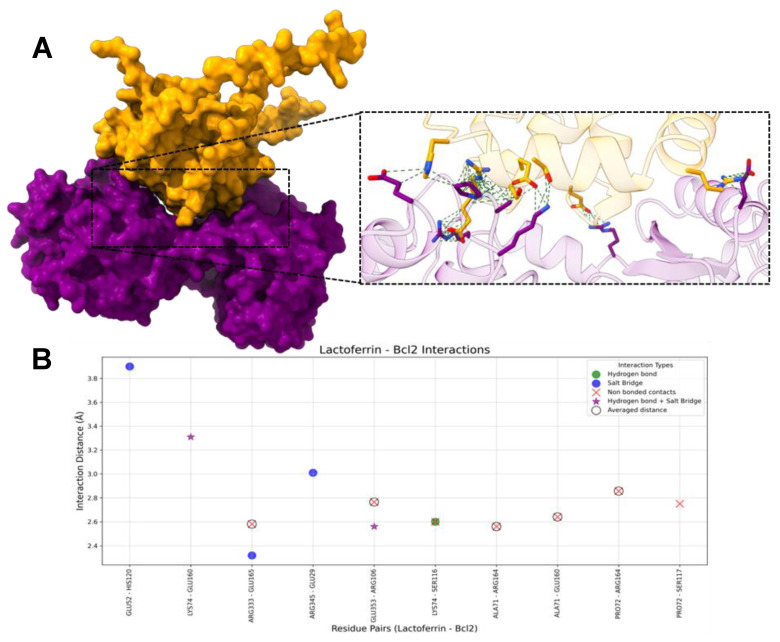
Interactions between hLf and Bcl2. (**A**) The molecular surface representation of the hLf-Bcl2 complex. HLf is purple and Bcl2 is yellow mustard, highlighting the interface where interactions occur. The zoomed-in view displays the residues involved in interacting in the complex, with green dashed lines representing contacts between residues. (**B**) The scatter plot illustrates interaction distances (Å) for residue pairs involved in Salt Bridges (blue), Hydrogen Bonds (green), Non-Bonded Contacts (red), and Hydrogen bonds + Salt Bridge (purple star). Residue pairs with averaged distances for “Non-Bonded Contacts” (i.e., residue pairs that presented more than one “Non-Bonded Contact” interaction) are marked with hollow black circles around red “X” markers. The *x*-axis represents hLf and Bcl2 residue pairs, and the *y*-axis represents the interaction distance in Å.

**Figure 5 ijms-26-02023-f005:**
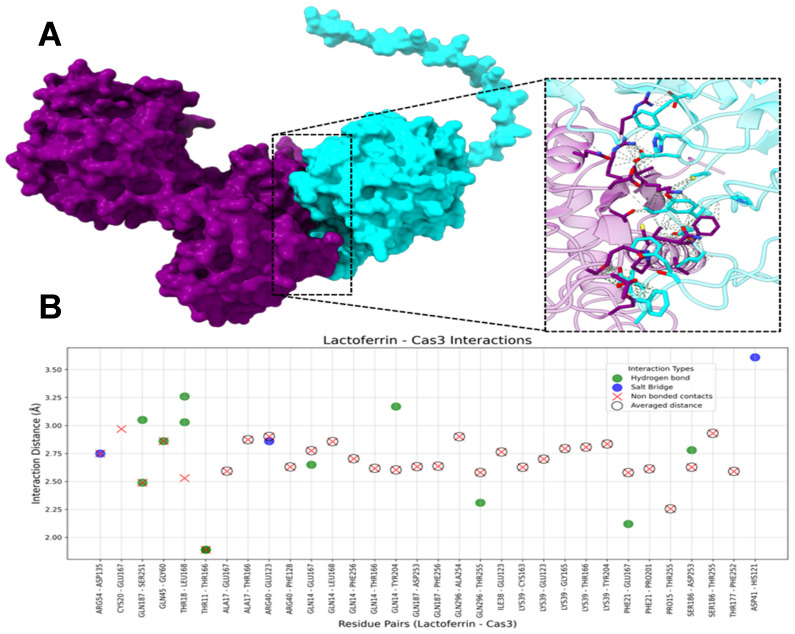
Interactions between hLf and Caspase 3 (Cas3). (**A**) The molecular surface representation of the hLf-Cas3 complex. HLf is purple and Caspase-3 is cyan, highlighting the interface where interactions occur. The zoomed-in view displays the residues involved in interacting in the complex, with green dashed lines representing contacts between residues. (**B**) The scatter plot illustrates interaction distances (Å) for residue pairs involved in Salt Bridges (blue), Hydrogen Bonds (green), and Non-Bonded Contacts (red). Residue pairs with averaged distances for “Non-Bonded Contacts” (i.e., residue pairs that presented more than one “Non-Bonded Contact” interaction) are marked with hollow black circles around red “X” markers. The *x*-axis represents hLf and Caspase-3 residue pairs, and the *y*-axis represents the interaction distance in Å.

**Figure 6 ijms-26-02023-f006:**
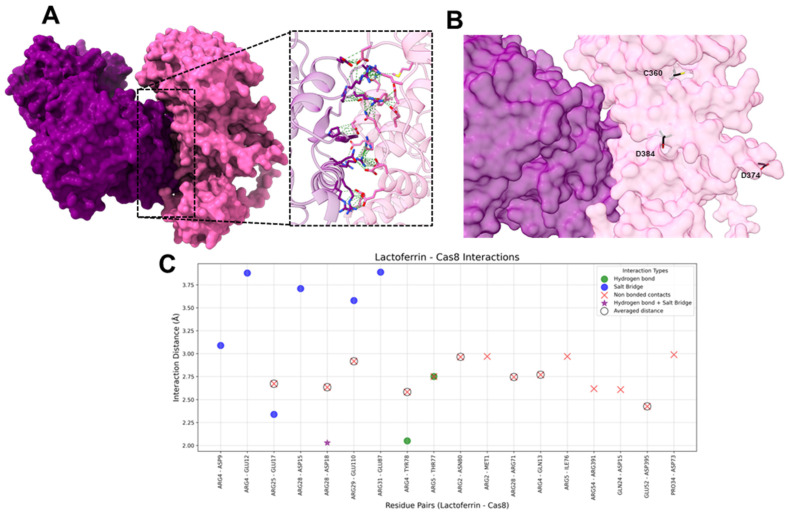
Interactions between hLf and Caspase-8. (**A**) The molecular surface representation of the hLf-Cas8 complex. HLf is purple and Caspase-8 is pink, highlighting the interface where interactions occur. The zoomed-in view displays the residues involved interacting in the complex, with green dashed lines representing contacts between residues. (**B**) The molecular surface representation of the hLf-Cas8 complex. HLf is shown in purple and Caspase-8 is shown in pink. Catalytic and functional residues of Cas8 are shown as sticks with carbon atoms in black, oxygen in red, and Sulphur in yellow. (**C**) The scatter plot illustrates interaction distances (Å) for residue pairs involved in Salt Bridges (blue), Hydrogen Bonds (green), Non-Bonded Contacts (red), and Hydrogen bonds + Salt Bridge (purple star). Residue pairs with averaged distances for “Non-Bonded Contacts” (i.e., residue pairs that presented more than one “Non-Bonded Contact” interaction) are marked with hollow black circles around red “X” markers. The *x*-axis represents hLf and Caspase-8 residue pairs, and the *y*-axis represents the interaction distance in Å.

**Figure 7 ijms-26-02023-f007:**
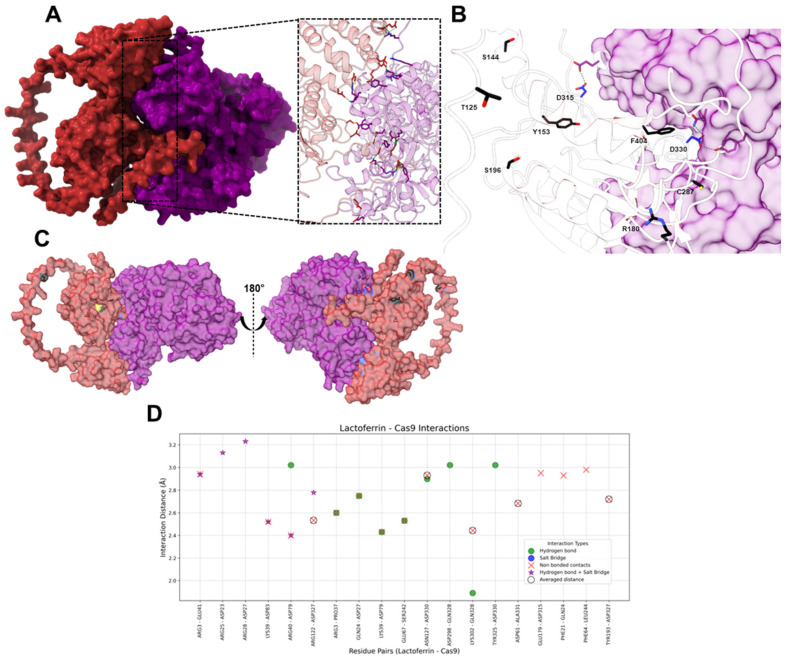
Interactions between hLf and Caspase-9. (**A**) The molecular surface representation of the hLf-Cas9 complex. HLf is in purple and Caspase-9 is in dark red, highlighting the interface where interactions occur. The zoomed-in view displays the residues involved interacting in the complex, with green dashed lines representing contacts between residues. (**B**) The molecular surface-ribbon representation of the hLf-Cas9 complex. HLf surface is shown in purple and Cas9 is shown as a ribbon, with functional and catalytic residues shown as sticks with carbon atoms in black/blue, oxygen in red, nitrogen in blue, and Sulphur in yellow. (**C**) The molecular surface representation of the hLf-Cas9 complex. HLf is shown in purple and Caspase-9 is shown in dark-red. Functional and catalytic residues are shown as sticks with carbon atoms in black, oxygen in red, nitrogen in blue, and Sulphur in yellow. (**D**) The scatter plot illustrates interaction distances (Å) for residue pairs involved in Salt Bridges (blue), Hydrogen Bonds (green), Non-Bonded Contacts (red), and Hydrogen bonds + Salt Bridge (purple star). Residue pairs with averaged distances for “Non-Bonded Contacts” (i.e., residue pairs that presented more than one “Non-Bonded Contact” interaction) are marked with hollow black circles around red “X” markers. The *x*-axis represents hLf and Caspase-9 residue pairs, and the *y*-axis represents the interaction distance in Å.

**Figure 8 ijms-26-02023-f008:**
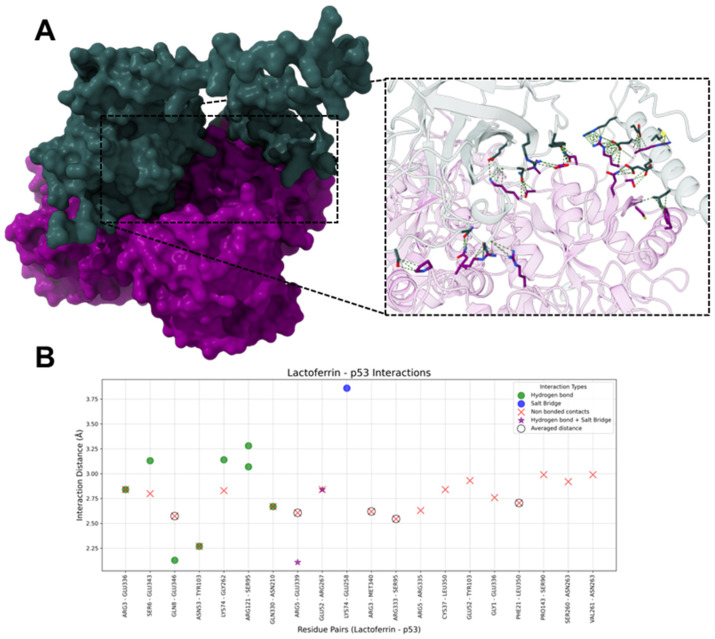
Interactions between human Lactoferrin (hLf) and p53. (**A**) The molecular surface representation of the hLf-p53 complex. HLf is purple and p53 is blue-gray, highlighting the interface where interactions occur. The zoomed-in view displays the residues involved interacting in the complex, with green dashed lines representing contacts between residues. (**B**) The scatter plot illustrates interaction distances (Å) for residue pairs involved in Salt Bridges (blue), Hydrogen Bonds (green), Non-Bonded Contacts (red), and Hydrogen bonds + Salt Bridge (purple star). Residue pairs with averaged distances for “Non-Bonded Contacts” (i.e., residue pairs that presented more than one “Non-Bonded Contact” interaction) are marked with hollow black circles around red “X” markers. The *x*-axis represents hLf and p53 residue pairs, and the *y*-axis represents the interaction distance in Å.

**Figure 9 ijms-26-02023-f009:**
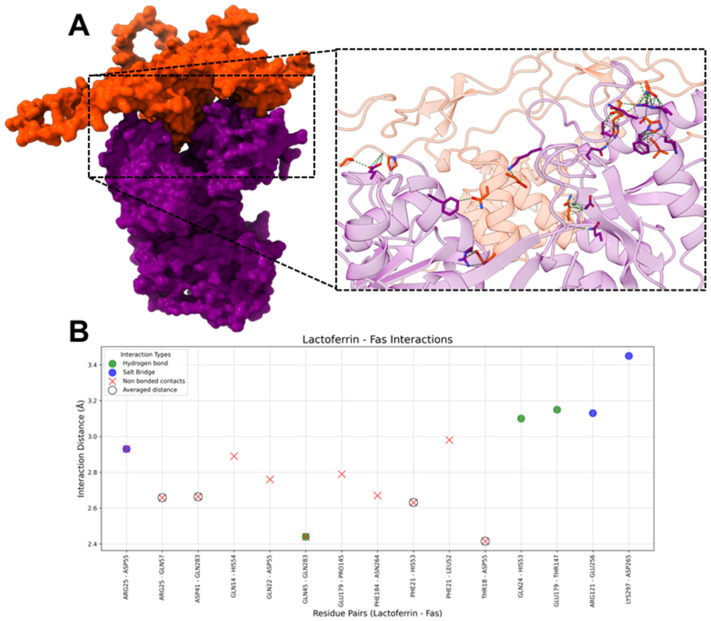
Interactions between hLf and Fas. (**A**) The molecular surface representation of the hLf-Fas complex. HLf is in purple and Fas is in orange, highlighting the interface where interactions occur. The zoomed-in view displays the residues involved interacting in the complex, with green dashed lines representing contacts between residues. (**B**) The scatter plot illustrates interaction distances (Å) for residue pairs involved in Salt Bridges (blue), Hydrogen Bonds (green), and Non-Bonded Contacts (red). Residue pairs with averaged distances for “Non-Bonded Contacts” (i.e., residue pairs that presented more than one “Non-Bonded Contact” interaction) are marked with hollow black circles around red “X” markers. The *x*-axis represents hLf and Fas residue pairs, and the *y*-axis represents the interaction distance in Å.

**Figure 10 ijms-26-02023-f010:**
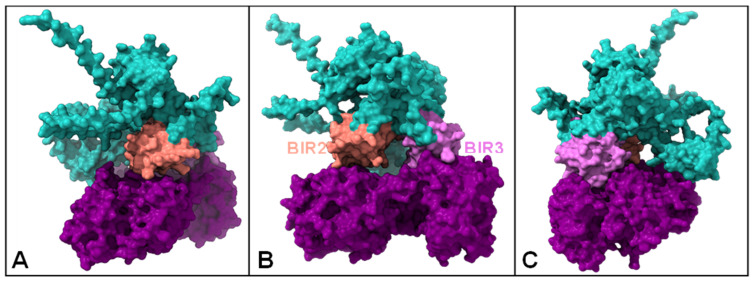
Interactions between hLf and XIAP. (**A**–**C**) The molecular surface representation of the hLf-XIAP complex. HLf is in purple, XIAP in blue, the BIR2 domain in salmon, and the BIR3 domain in pink.

**Figure 11 ijms-26-02023-f011:**
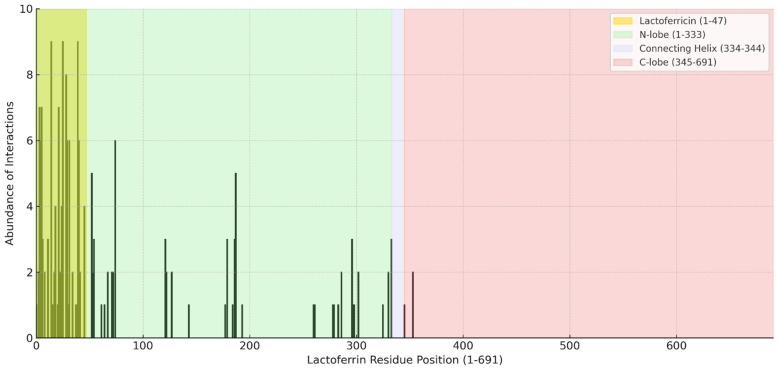
Residue-Specific Interaction Profile of hLf with Apoptotic Proteins: Highlighting the Predominant Role of Lactoferricin (Lfcin). Total interactions observed in docking results between hLf and apoptotic proteins across the entire Lfcin sequence. The *x*-axis represents the residue position of hLf (1–691), while the *y*-axis indicates the abundance of interactions. The structural regions of hLf are highlighted, including Lfcin (1–47), N-lobe (1–333), Connecting Helix (334–344), and C-lobe (345–691). The results reveal that the Lfcin region, a key segment of the N-lobe, exhibits the highest interaction frequency with apoptotic proteins, suggesting its critical role in mediating protein-protein interactions.

**Figure 12 ijms-26-02023-f012:**
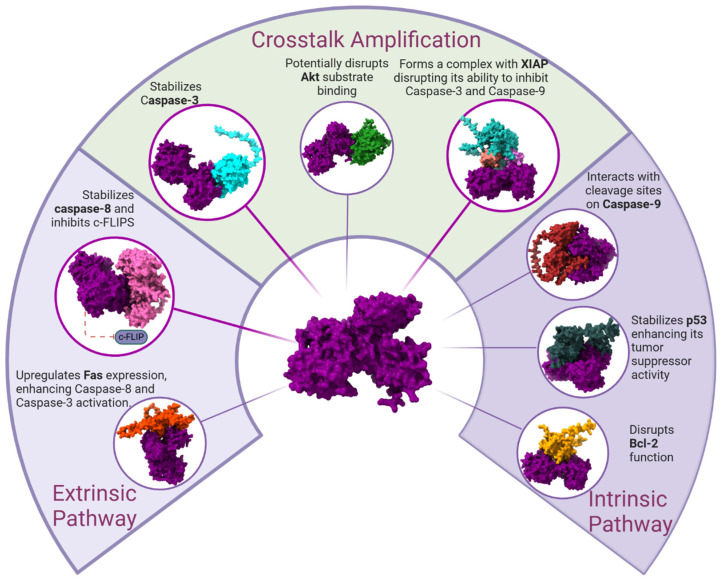
Molecular Docking Analysis of human Lactoferrin Interactions with Apoptotic Pathway Proteins. The image summarizes the docking results between human Lactoferrin (hLf) and key apoptotic proteins, highlighting its potential role in apoptosis regulation. It illustrates interactions within both the extrinsic and intrinsic apoptotic pathways, as well as its contribution to crosstalk amplification. In the extrinsic pathway, hLf upregulates Fas expression, enhancing Caspase-8 and Caspase-3 activation, while also stabilizing Caspase-8 and inhibiting c-FLIPS. In the intrinsic pathway, hLf stabilizes p53, enhances its tumor suppressor activity, disrupts Bcl-2 function, and interacts with cleavage sites on Caspase-9, promoting apoptosis. Additionally, in the crosstalk amplification mechanism, hLf potentially disrupts Akt substrate binding, forms a complex with XIAP, interfering with its ability to inhibit Caspase-3 and Caspase-9, and stabilizes Caspase-3. These interactions suggest that hLf may promote apoptosis by enhancing caspase activation, interfering with survival signals, and modulating key apoptotic regulators, providing insights into its potential anticancer effects.

**Table 1 ijms-26-02023-t001:** Apoptotic Proteins: Functions and Cancer Implications.

Protein	Functions in Apoptosis	Cancer Implications	References
Akt	Anti-apoptotic protein. Stimulates cell proliferation survival, alters gene transcription and inhibits glycogen synthase kinase (GSK), resulting in enhanced glucose metabolism.	*AKT1* gene mutation E17K may serve as an oncogene observed in breast, colorectal, lung, and ovarian cancers. PI3K-AKT signaling pathway is frequently mutated in cancer and is one of the most dysregulated signaling pathways in tumorigenesis.	[25,26,27,28,29]
Bcl-2	Anti-apoptotic protein. Binds to pro-apoptotic family members, sequestering them and thereby preventing the release of cytochrome c and other pro-apoptotic factors from the mitochondria thus inhibiting cell death.	Bcl-2 is elevated in cancers like prostate, lung, colorectal, gastric, renal, neuroblastoma, non-Hodgkin’s lymphoma, and leukemia. *Bcl-2* overexpression is indicative of cancer promotion and metastasis. Bcl-2 plays an important role in resistance of cancer cells to chemotherapy or radiation therapy	[18,30,31,32,33,34,35]
p53	Tumor suppressor. Translocates from the cytosol to the nucleus, where it activates the expression of pro-apoptotic and autophagic genes. Additionally, p53 can translocate to the mitochondrial membrane, where it regulates MOMP.	*TP53* gene is one of the most frequently mutated genes in cancers, including breast, colorectal, lung, ovarian, and glioblastoma. Mutations disrupt p53’s DNA-binding domain, preventing apoptosis activation and promoting tumor progression.	[8,36,37]
Fas	Death receptor. Activates the extrinsic apoptotic pathway via DISC (Death-Inducing Signaling Complex) formation upon Fas ligand binding.	Several studies have identified Fas as a positive prognostic marker in cancer. Some cancers demonstrate reduced cell surface levels of Fas and thus evade the control system via ligand-induced apoptosis	[38,39,40]
Caspase-3	The major executioner caspase. Cleaves cellular proteins, leading to cell dismantling during apoptosis.	The reduction of Caspase-3 in various cancer cells disrupts apoptosis, contributing to their survival and progression. Caspase-3 (*CASP3*) is linked to cancer through somatic mutations, although these mutations are not highly prevalent. XIAP inhibits Caspase-3, leading to apoptosis resistance and promoting tumor progression.	[41,42,43,44,45]
Caspase-8	Initiator caspase in the extrinsic pathway. Activated through death receptor signaling. Once activated, it triggers the activation of executioner caspases (Caspase-3 and -7).	Mutations, suppression or hypermethylation of Caspase-8 (*CASP8*) are linked to resistance to death receptor-mediated apoptosis in cancers, including gastric, lung, breast, pancreatic, and glioblastoma cancer. Restoring the expression of Caspase-8 (*CASP8*) in some cancers may lead to non-apoptotic roles like cell migration	[41,42,43,46,47]
Caspase-9	Initiator caspase in the intrinsic pathway. Activated by the apoptosome complex (cytochrome c, Apaf-1 (Apoptosis protease-activating factor-1)). Once activated, it triggers the activation of executioner caspases (Caspase-3 and -7).	Caspase-9 (*CASP9*) polymorphisms are significantly associated with the risk of lung, bladder, pancreatic, colorectal, and gastric cancers.	[41,48,49,50,51,52]
XIAP	Crucial protein that plays a multifaceted role in regulating cell death and immune responses, it belongs to the inhibitor of apoptosis protein (IAP) family, which comprises eight different proteins that share a zinc-binding baculovirus IAP-repeat (BIR) domain. It is the most potent inhibitor of caspases; it binds directly to and inhibits Caspases 3 and 9.	Elevated XIAP expression is found in various cancer types and is associated with poor prognosis and resistance to chemotherapy. XIAP contributes to cancer development by inhibiting apoptosis and promoting cell survival.	[53,54,55,56]

Summary of apoptotic proteins, their roles in apoptosis pathways, and their relevance in cancer.

## Data Availability

The molecular docking data generated and analyzed in this study are available from the corresponding author upon reasonable request.

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
