# Peer review of "Molecular Docking of Lactoferrin with Apoptosis-Related Proteins Insights into Its Anticancer Mechanism"

_ijms, 2025, doi:10.3390/ijms26052023_

Round 1

Reviewer 1 Report

Comments and Suggestions for Authors

This is a nice in silico structural biology study of lactoferrin, harnassing the power of the new AlphaFold3 software to explore interesting and potentially important interactions between lactoferrin and apoposis-related proteins. The binding partners to explore are well-chosen to be interesting and potentially important. The analysis of the resulting molecular structures is well done. Now that AlphaFold is available, we should all take advantage of the AlphaFold breakthrough and do studies like this. This study is a good example to others of what to do and how to analysis the resulting models. I think that this manuscript is worth publishing as is. The manuscript is very nicely written. The introduction is clear and helpful and segues into the in-silico experiments. It is clear what the researchers are doing and why. The molecular images look great.

This manuscript does not contain any experimental evidence to support or refute the resulting models, which would require quite different skills, would require orders of magnitude more resources, and would provide results that are so much more valuable. It is appropriate that AlphaFold be used for many in silico studies, such as this well executed study which takes advantage of the latest available, superior version of AlphaFold 3.

Now that we are learning how to use AlphaFold, I wonder if there are further in silico steps we could take to help validate models. Perhaps produce a Ramachandran Plot and verify independently of AlphaFold that torsion angles are correct and that there are no steric clashes? I wonder what in silico negative control experiments we could do, but I can't think of any. Are there any proteins that globular like lactoferrin and are known to NOT binding the binding partners resulting from this study or are there any proteins that seem similar to the binding partners and it is known that they DO NOT bind lactoferrin? If so, one could try Alphafold binding of the 2 proteins knowns to not bind, and demonstrate that AlphaFold does not bind them together.

In lines 181 and 724, why do you call the Alphafold models "homology models"? My understanding is that when you take an existing model of a 3D-folded amino acid sequence, and then make changes to the sequence (eg. cancer mutations or using the similar sequence of another species) and use the existing model to 3D model the new sequence, then that is a homology model.

In Figure 3B, the x-axis is unreadable, so I don't really know what residues I'm looking at. The y-axis and legend are almost unreadable. Same for Figure B of the other figures. The plots are a good idea and worth presenting.

Author Response

Comment 1:

"Now that we are learning how to use AlphaFold, I wonder if there are further in silico steps we could take to help validate models. Perhaps produce a Ramachandran Plot and verify independently of AlphaFold that torsion angles are correct and that there are no steric clashes?"

Response 1:

Thank you for pointing this out. We appreciate your suggestion. Unlike homology-modeled structures, which require Ramachandran plot validation, AlphaFold 3 integrates steric, torsional, and conformational constraints directly into its predictions. Therefore, Ramachandran plots provide limited additional insight for AlphaFold 3-generated models. Instead, we relied on pLDDT, pTM, ipTM, and PAE metrics, which have been demonstrated to correlate well with experimentally validated structures. These metrics offer a robust validation approach tailored to AlphaFold-generated structures.

 Comment 2:

"I wonder what in silico negative control experiments we could do, but I can't think of any. Are there any proteins that globular like lactoferrin and are known to NOT binding the binding partners resulting from this study or are there any proteins that seem similar to the binding partners and it is known that they DO NOT bind lactoferrin?"

 Response 2:

We appreciate this insightful suggestion. We performed a negative control docking experiment, which is included in the manuscript. Additional details, including supporting figures and tables, are provided in the Supplementary Material.

 Comment 3:

"In lines 181 and 724, why do you call the AlphaFold models 'homology models'?"

Response 3:

Thank you for your observation. We have replaced "homology models" with "predicted structures" throughout the manuscript to ensure accuracy in terminology.

 Comment 4:

"In Figure 3B, the x-axis is unreadable, so I don't really know what residues I'm looking at. The y-axis and legend are almost unreadable. Same for Figure B of the other figures. The plots are a good idea and worth presenting."

Response 4:

We appreciate this feedback. We have improved the readability of all figures, ensuring that axis labels and legends are now clear and legible.

 Thank you again for your thoughtful review and valuable suggestions, which have helped improve the clarity and accuracy of our manuscript.

Reviewer 2 Report

Comments and Suggestions for Authors

This is a very well written and informative paper. The results are clearly presented and argued.

The manuscript could be improved slightly by:

Reword incomplete sentence: BAX, 76 regulated by the tumor suppressor gene p53, and BAK oligomerize upon activation.

(MOMP) is redefined each time it is used. Rather leave the abbreviation out.

The experimental section is very brief and only contains a list of the programs used. There are no details like degrees of freedom etc used in the docking. How many docking poses were obtained?  One needs to know this to understand the comments like “Though less frequent, salt bridges played a significant role in electrostatic stabilization.”  If there is only one docked pose then there is either an interaction or not – it cannot be averaged as was done here.

Author Response

We sincerely appreciate your thoughtful comments and suggestions, which have helped improve the clarity and completeness of our manuscript. Below, we address each of your points and outline the corresponding revisions made.

Comment 1:

"Reword incomplete sentence: BAX, 76 regulated by the tumor suppressor gene p53, and BAK oligomerize upon activation."

Response:

Thank you for pointing this out. We have reworded the sentence in the revised manuscript.

Comment 2:

"(MOMP) is redefined each time it is used. Rather leave the abbreviation out."

Response:

We appreciate this suggestion. We have removed redundant definitions of MOMP and now use only the abbreviation throughout the text for consistency.

Comment 3:

"The experimental section is very brief and only contains a list of the programs used. There are no details like degrees of freedom, etc., used in the docking. How many docking poses were obtained? One needs to know this to understand the comments like 'Though less frequent, salt bridges played a significant role in electrostatic stabilization.' If there is only one docked pose, then there is either an interaction or not – it cannot be averaged as was done here."

Response:

Thank you for your valuable feedback. We have expanded the methodology section to provide more details on the structure prediction and docking process.

Additionally, we have clarified the role of salt bridges for better comprehension in the manuscript.

We appreciate your insight, and these clarifications have been incorporated into the revised manuscript. Thank you again for your helpful suggestions.

Reviewer 3 Report

Comments and Suggestions for Authors

I have reviewed the manuscript titled "Molecular Docking of Lactoferrin with Apoptosis-Related 2 Proteins Insights into its Anticancer Mechanism" by Lidia et al. The article discusses apoptosis regulation via studying the interaction between human lactoferrin (hLf) and XIAP and Caspase-3, Fas, Bcl-2, and Akt using docking studies. I appreciate the main idea of the paper and certain aspects, such as the selection of apoptotic proteins. However, there are some areas that require improvement, and my comments are as follows:

  1.    The introduction lacks sufficient and strong rationale behind the study. It is ver long and should be reduced. The table can be transferred to SI.
  2.    Authors are suggested to perform simulation studies to estimate the stability of the complexes, without which the results are incomplete.
  3.    Figure 12: This should be discussed separately before the conclusion.
  4.    Authors are also suggested to identify and discuss the limitations of the present study.

Author Response

We sincerely appreciate your thoughtful comments and suggestions, which have helped improve the clarity and structure of our manuscript. Below, we address each of your points and outline the corresponding revisions made.

Comment 1:

"The introduction lacks sufficient and strong rationale behind the study. It is very long and should be reduced. The table can be transferred to SI."

Response:

Thank you for your suggestion. We have modified the introduction to enhance clarity and reduce its length while ensuring the rationale behind the study is clearly conveyed. However, we consider Table 1 an essential part of the introduction, as it provides critical background information relevant to the study. Therefore, we have retained it in the main text rather than moving it to the SI.

Comment 2:

"Authors are suggested to perform simulation studies to estimate the stability of the complexes, without which the results are incomplete."

Response:

We acknowledge the importance of molecular dynamics (MD) simulations in evaluating complex stability. However, this study focuses on structure prediction and docking using AlphaFold 3, which provides reliable confidence metrics such as pLDDT, pTM, ipTM, and PAE for model assessment. Additionally, as suggested by one of the reviewers, we performed a negative control docking experiment, which has been included in the manuscript. This additional analysis enhances the confidence in our study by validating the specificity of the predicted interactions.

While MD simulations would provide further insights, they are currently beyond our possibilities due to a lack of the necessary computational resources and equipment. Nevertheless, we highlight MD simulations as a potential future direction in the revised discussion.

Comment 3:

"Figure 12: This should be discussed separately before the conclusion."

Response:

Thank you for this suggestion. We have moved the discussion of Figure 12 to the Discussion section and removed it from the Conclusion to improve the logical flow of the manuscript.

Comment 4:

"Authors are also suggested to identify and discuss the limitations of the present study."

Response:

We appreciate this valuable suggestion. We have included a section discussing the limitations of the study, addressing aspects such as the lack of experimental validation, the absence of MD simulations, and the inherent constraints of in silico docking predictions. This addition provides a more balanced interpretation of our findings.

We sincerely appreciate your constructive feedback, which has helped us refine the manuscript. Thank you again for your thoughtful review.
